# Estimation of Cosmic-Ray-Induced Atmospheric Ionization and Radiation at Commercial Aviation Flight Altitudes

Panagiota Makrantoni [1], Anastasia Tezari [1,2], Argyris N. Stassinakis [1], Pavlos Paschalis [1], Maria Gerontidou [1], Pantelis Karaiskos [3], Alexandros G. Georgakilas [4], Helen Mavromichalaki [1,*], Ilya G. Usoskin [5], Norma Crosby [6] and Mark Dierckxsens [6]

[1] Athens Cosmic Ray Group, Faculty of Physics, National and Kapodistrian University of Athens, 15784 Athens, Greece; pmakrantoni@phys.uoa.gr (P.M.); anatez@med.uoa.gr (A.T.); a-stasinakis@phys.uoa.gr (A.N.S.); ppaschalis@phys.uoa.gr (P.P.); mgeront@phys.uoa.gr (M.G.)

[2] Eugenides Foundation, 17564 Athens, Greece

[3] Medical Physics Laboratory, Faculty of Medicine, National and Kapodistrian University of Athens, 11517 Athens, Greece; pkaraisk@med.uoa.gr

[4] DNA Damage Laboratory, Physics Department, School of Applied Mathematical and Physical Sciences, National Technical University of Athens (NTUA), Zografou, Athens 15780, Greece; alexg@mail.ntua.gr

[5] Space Physics and Astronomy Research Unit and Sodankylä Geophysical Observatory, University of Oulu, FIN-90014 Oulu, Finland; ilya.usoskin@oulu.fi

[6] Royal Belgian Institute for Space Aeronomy, 1180 Brussels, Belgium; norma.crosby@aeronomie.be (N.C.); mark.dierckxsens@aeronomie.be (M.D.)

\* Correspondence: emavromi@phys.uoa.gr

**Abstract:** The main source of the ionization of the Earth's atmosphere is the cosmic radiation that depends on solar activity as well as geomagnetic activity. Galactic cosmic rays constitute a permanent radiation background and contribute significantly to the radiation exposure inside the atmosphere. In this work, the cosmic-ray-induced ionization of the Earth's atmosphere, due to both solar and galactic cosmic radiation during the recent solar cycles 23 (1996–2008) and 24 (2008–2019), was studied globally. Estimations of the ionization were based on the CRAC:CRII model by the University of Oulu. The use of this model allowed for extensive calculations from the Earth's surface (atmospheric depth 1033 g/cm$^2$) to the upper limit of the atmosphere (atmospheric depth 0 g/cm$^2$). Monte Carlo simulations were performed for the estimation quantities of radiobiological interest with the validated software DYASTIMA/DYASTIMA-R. This study was focused on specific altitudes of interest, such as the common flight levels used by commercial aviation.

**Keywords:** cosmic rays; ionization; radiation; atmosphere; solar cycle; flight level; aviation

## 1. Introduction

Cosmic rays are highly energetic particles of extraterrestrial origin. There are two main components of cosmic rays: galactic cosmic rays (GCRs), which originate from outside of our Solar System and Solar Energetic Particles (SEPs), which are accelerated during eruptive processes on the Sun [1]. As cosmic rays travel through the interplanetary space and reach the terrestrial atmosphere, (these rays are named primary cosmic rays), they penetrate by colliding with nuclei of atoms and ions of the atmosphere, thus creating nucleonic, muonic and electromagnetic cascades named secondary cosmic rays, as the primary particles are absorbed inside the atmosphere due to ionization losses. In this way, cosmic rays affect the physical–chemical properties of the atmosphere, i.e., its ion balance, [2,3] and may even affect the regional climate's variability [4]. The Earth's magnetic field acts as a charge discriminator and modulates the cosmic ray flux that reaches each location on the Earth.

Since cosmic rays are always present as a natural radiation background, they constitute a major factor in the ionization of the atmosphere. This process is called cosmic-ray-induced ionization (CRII). The GCRs affect the CRII by following an 11-year modulation oppositely

correlated to the solar activity, i.e., the greater the solar activity, the lower the intensity of the CRII is. On the other hand, strong fluxes of the unpredictable SEPs produced in solar flares or coronal mass ejections (CMEs) most likely occur during periods of intense solar activity and mostly affect the polar regions and high altitudes, where the magnetic field lines are open and the energetic particles may deposit their energy, even at 20 km a.s.l. It is noteworthy that GCRs are referred to as the continuous flux of the charged particles which originate from different sources within the intergalactic space, while SEPs make up the solar component of cosmic rays, associated with an increase in particle fluxes released in the interplanetary space after great solar activity. SEPs also create hazards for satellites, spacecraft, high-altitude aircraft, as well as for the health of air crews and space crews, due to the enhanced radiation environment SEPs create [5,6].

As cosmic rays contribute to the production of ion pairs, which are involved in several atmospheric processes, numerous studies indicate that ionization induced by CRs may affect different climate parameters [7,8] and so the computation of CRII is considered necessary. Specifically, the atmospheric ionization may alter the physical and chemical properties of the atmosphere and affect several processes, such as aerosol and cloud formation, atmospheric transparency, cloud cover, cyclogenesis and precipitation, especially in regions of middle and high geographic latitudes. Therefore, several numerical models were created and validated via comparison with direct observations and measurements of the CRII, e.g., the Sofia model [8,9], the Bern model, also called ATMOCOSMIC [10,11] and the Oulu model, also called CRAC:CRII [12,13]. Results from the latter model [14] are used in this work.

Aside from cosmic rays affecting the composition of the atmosphere and contributing to climate configuration, they may also affect human health [15,16]. The way that CRII is modulated and distributed inside the atmosphere affects human exposure to radiation, suggesting that air crew members and frequent flyers of commercial flights should be treated in a specific way and extra safety measures and necessary regulations should be applied during their flights. Other than ones referring to the general public, specific regulations and safety measures do not yet exist for frequent flyers. However, the European Commission, as well as other entities such as the International Commission on Radiation Units and Measurements (ICRU) and the International Committee on Radiological Protection (ICRP), have adopted a series of recommendations and frameworks regarding the determination of the occupational exposure of aviation crews to cosmic radiation, as well as the most efficient measures and counteractions to ensure radiation protection [17–19]. For this reason, several studies on the problem have been performed [20–27], while various models and tools have also been developed by the scientific community in cooperation with the aviation industry. Some of these well-known models are the following: SIEVERT [28], AVIDOS [29], NAIRAS [30], CARI [31], CALVADOS [32] SPENVIS [33], CRAC:DOSE [23] and PLANETOCOSMICS [10].

Furthermore, ionization also affects the avionic electronic systems during a flight, with single event effects (SEEs) being a main factor in this [34,35]. To expand, a single secondary high-energy atmospheric neutron can collide with a nucleus of the semiconductor, causing an ionization charge that can affect a semiconductor device. The most common effects of SEEs are soft errors, firm errors and hard errors that decrease the performance and the availability of electronic systems. Furthermore, radiation can also affect optical components (i.e., LEDs, lasers and optical fibers), by changing their optical properties and causing displacement damage. Usually, the electronic and optoelectronic devices anneal after the irradiation has stopped but, in some cases, radiation can cause permanent damage [36–38]. Therefore, it is crucial to estimate the ionization and radiation levels during a flight in order to be able to maintain reliability standards.

In this work, the CRII was calculated for three different flight levels (FLs): FL310 (9.45 km a.s.l. or 31,000 ft), FL350 (10.67 km a.s.l. or 35,000 ft) and FL390 (11.89 km a.s.l. or 39,000 ft). The model used for the aforementioned calculations was the CRAC:CRII model in its extended version [12,13]. The CRII for these typical FLs is depicted in ionization

maps (Figure 1), showing the distribution of CRII globally for specific phases of solar cycles 23 and 24 (i.e., solar maxima and minima). More to that, time profiles of the monthly distribution of CRII for selected magnetic cut-off rigidities for the same FLs during solar cycles 23 and 24 will be presented.

In addition, a similar study concerning the radiation assessment of occupational exposure to cosmic rays, specifically the estimation of the ambient equivalent dose rate for the typical FLs, was performed. For these calculations, the software application Dynamic Atmospheric Shower Tracking Interactive Model Application (DYASTIMA) [39,40] of the Athens Cosmic Ray Group was used. Finally, regarding the investigation in this work, a combined study was performed and a correlation between these two physical parameters is shown.

## 2. Technical Analysis and Data Selection

For the ionization induced by cosmic rays, the CRAC:CRII model of the Oulu University was used—a numerical model that computes the CRII from the sea level to up to 40 km in the atmosphere, for every location on Earth. This model uses the Monte Carlo CORSIKA tool (v.6.617 August 2007) [41], which provides a full development simulation of an electromagnetic–muon–nucleonic cascade in the atmosphere, as well as the FLUKA package for the low-energy interactions (v.2006.3b March 2007) [42] and is fully described in [12,13].

Moreover, concerning calculations of the CRII for specific latitudes, altitudes and time periods, considering both solar and galactic cosmic rays, the "Cosmic Ray Induced Ionization: Do-it-yourself kit" (http://cosmicrays.oulu.fi/CRII/CRII.html) (accessed on 5 May 2022) of the Oulu Cosmic Ray Station was used, and the monthly and annual values of the modulation parameter Phi (in MV), reconstructed from the ground-based cosmic ray data, are provided [43,44]. The modulation parameter corresponds to the local interstellar spectrum (LIS) of cosmic rays, as provided by [45].

In order to calculate the ambient dose equivalent rate dH*(10)/dt, DYASTIMA was used [39]. Monte Carlo simulations of the secondary cascades taking place in the different atmospheric layers were performed with this independent GEANT4 software tool [46–48], which allowed for the determination of several characteristics of the cascade, such as the energy of the particles and the energy deposits at the different atmospheric layers. The FTFP_BERT_HP GEANT4 physics list was used, as it adequately describes all processes taking place due to secondary cascades. Then, several radiobiological quantities were calculated with the DYASTIMA-R extension. Specifically, the operational quantity dH*(10)/dt was estimated by taking into consideration the different radiation weighting factors that corresponded to the different types of secondary cosmic ray particles [9]. DYASTIMA/DYASTIMA-R is a validated tool [40,49,50], as it meets the criteria provided by the ICRU and ICRP documents [18,19] regarding radiation protection in the aviation sector. DYASTIMA software was provided through the portal of the Athens Neutron Monitor Station (A.Ne.Mo.S.) (http://cosray.phys.uoa.gr/index.php/dyastima) (accessed on 6 April 2022), while a database of selected simulated scenarios is available as a federated product on the ESA SWE Portal (https://swe.ssa.esa.int/dyastima-federated) (accessed on 6 April 2022).

The required input parameters for performing a simulation with DYASTIMA concern the characteristics of the planet and its atmosphere, as well as the differential spectrum of the incoming primary cosmic ray particles at the top of the atmosphere. As far as the simulations presented in this work are concerned, the atmospheric profile was based on the International Standard Atmosphere (ISA) model [51], while the ISO15390 model was used for the determination of the primary cosmic ray spectra [52]. To take into account the effect of the geomagnetic field, maps of the cut-off rigidity threshold values as a function of the geographic coordinates were used, based on the International Geomagnetic Reference Field (IGRF) [53–56]. The magnetic field components were obtained via the National Oceanic and

Atmospheric Administration portal (https://www.ngdc.noaa.gov/geomag/) (accessed on 24 March 2022).

## 3. Results

In this work, a study of cosmic-ray-induced ionization, computed via the CRAC:CRII model [12,13], along with the estimated ambient dose equivalent rate computed via the validated software DYASTIMA/DYASTIMA-R [39,40], was performed globally during the last two solar cycles (23 and 24) and focused on specific altitudes that corresponded to the most common commercial flight levels: FL310 (9.45 km a.s.l.), FL350 (10.67 km a.s.l.) and FL390 (11.89 km a.s.l.).

More specifically, the CRII at FL390 during the solar minima and solar maxima of solar cycles 23 and 24 is presented in Figure 1, globally, via ionization maps. Figure 1a depicts the CRII map that corresponds to the minimum of solar cycle 23 (in the year 1996), Figure 1b depicts the CRII map that corresponds to the maximum of solar cycle 23 (in the year 2001), Figure 1c depicts the CRII map that corresponds to the minimum of solar cycle 24 (in the year 2009) and Figure 1d depicts the CRII map that corresponds to the maximum of solar cycle 24 (in the year 2014). Comparing these four maps, it is clear that the ionization rate during the solar minima was greater than the ionization rate during the solar maxima of both cycles. This is due to the fact that the CRII followed the behavior of the cosmic ray intensity and was positively correlated with them, while it negatively correlated with the solar activity. In other words, the greater the solar activity, the lower the intensity of the CRII is [57–59].

Moreover, when comparing the solar minima and maxima of solar cycles 23 and 24, it is obvious that the CRII had greater values during solar cycle 24 than that of solar cycle 23, which was well expected, since solar cycle 24 is characterized as a relatively quiet solar cycle, unlike solar cycle 23, where the solar activity was greater. The minimum and maximum values for CRII and dH*(10)/dt obtained during this work for these specific time periods are presented in Table 1. Regarding the geographic coordinates, it was observed that, globally, the maximum ionization rate was found in polar regions while, at lower latitudes, the ionization rate reached minimum. This was due to the magnetic field of the Earth and the geomagnetic cut-off rigidity (Rc) that corresponded to each location, from 0 GV in polar regions to up to 17 GV in equatorial regions. The lower the geomagnetic cut-off rigidity (Rc), the more cosmic rays penetrated the magnetosphere and the atmosphere of the Earth; the CRs then ionized the atmosphere and created various effects [51]. Both the CRII and estimated ambient dose equivalent rate maps were generated based on the rigidity map of [53–56].

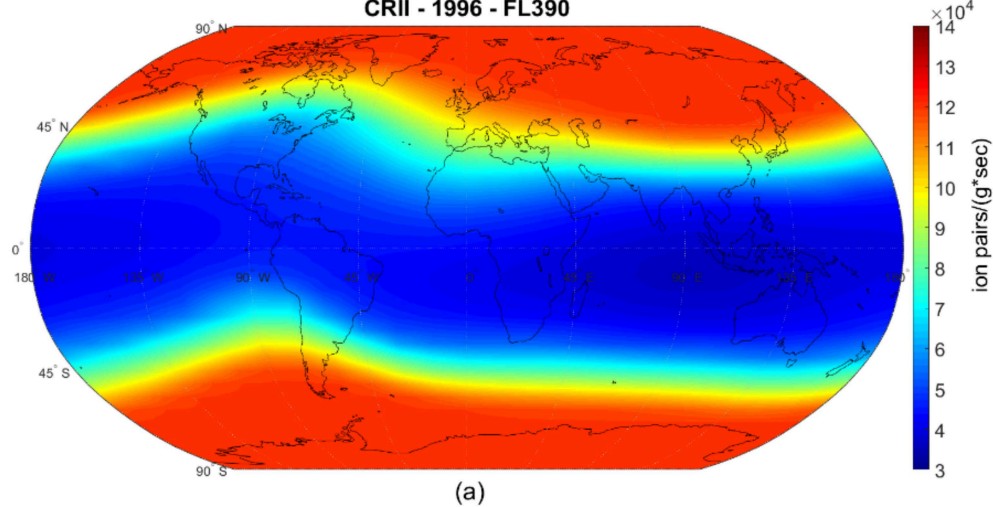

**Figure 1.** *Cont.*

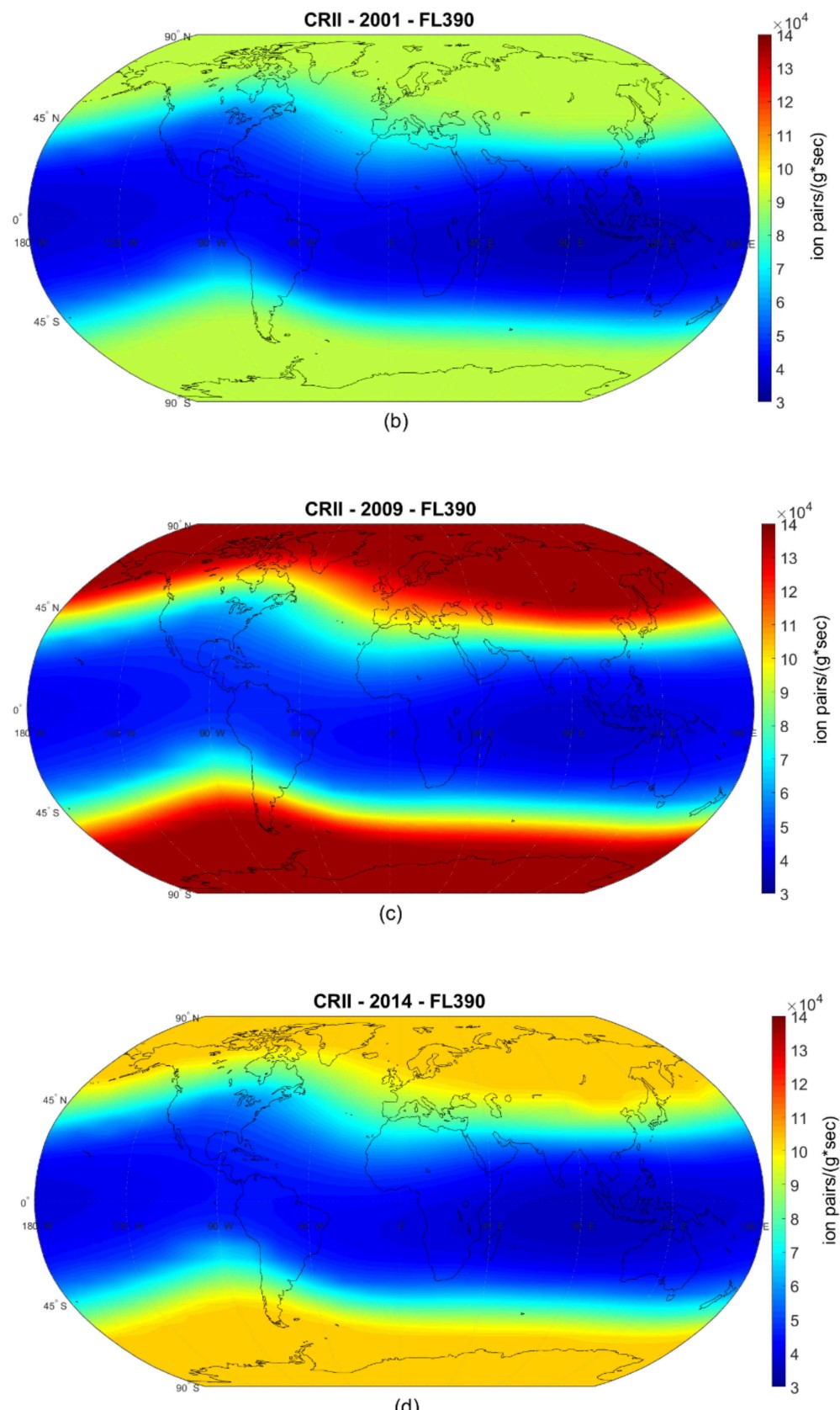

**Figure 1.** Maps of the CRII rate (ion pairs/(g*s)) at FL390: (**a**) during the minimum of solar cycle 23; (**b**) during the maximum of solar cycle 23; (**c**) during the minimum of solar cycle 24; (**d**) during the maximum of solar cycle 24.

**Table 1.** Minimum and maximum values of CRII and the estimated ambient dose equivalent rate during the minimum and maximum of solar cycles 23 and 24 at FL390.

| YEARS | CRII (Ion Pairs/(g*s)) | | dH*(10)/dt (µSv/h) | |
|---|---|---|---|---|
| | Minimum Values ×10⁴ | Maximum Values ×10⁴ | Minimum Values | Maximum Values |
| 1996 (SC23 min) | 3.6 | 12.0 | 1.22 | 6.83 |
| 2009 (SC24 min) | 3.7 | 13.6 | 1.24 | 7.05 |
| 2001 (SC23 max) | 3.4 | 9.1 | 1.18 | 5.49 |
| 2014 (SC24 max) | 3.5 | 11.0 | 1.19 | 5.50 |

With regard to the radiation exposure, the ambient dose equivalent rate at FL390 during the solar minima and solar maxima of solar cycles 23 and 24 is presented in Figure 2. A behavior similar to that of CRII was noticed. Greater values of the dH*(10)/dt were observed in the polar regions (Rc = 0–2 GV) and lower values near the equator (Rc = 15–17 GV), for both solar minima and maxima conditions. This was due to the dependence of the radiation levels at the atmospheric layers on the cosmic ray intensity [40,60]. As expected, the radiation exposure was greater during the solar minima compared to the solar maxima, due to the negative correlation between the solar activity and the intensity of the incoming cosmic ray particles. Greater values of dH*(10)/dt were also observed during the extended solar minimum in 2009 for both polar and equatorial regions, as compared to those observed during 1996. The observed differences can be characterized as relatively small, since the primary spectrum model used as input for the respective computations provided the estimation of the galactic component and did not take into account any SEPs which took place during this time period.

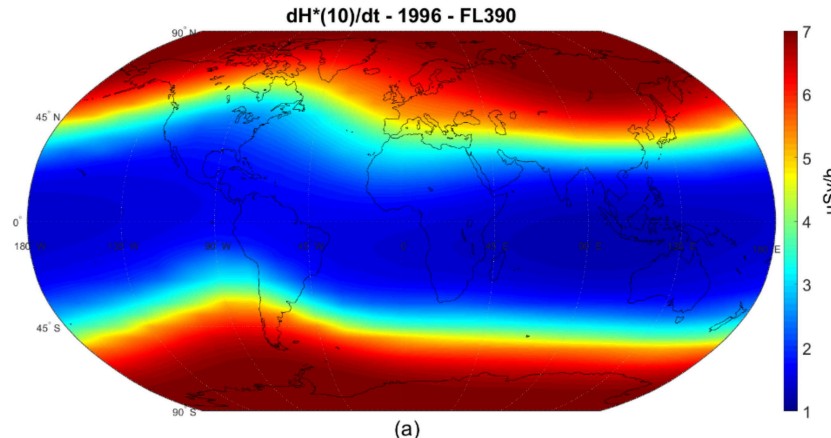

(a)

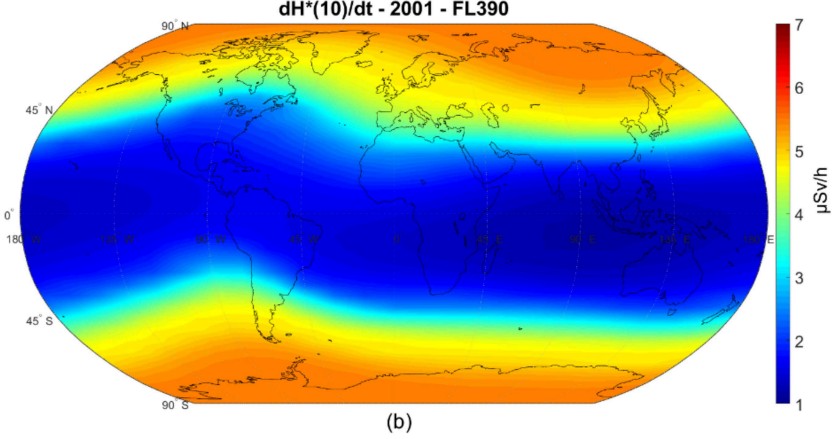

(b)

**Figure 2.** *Cont.*

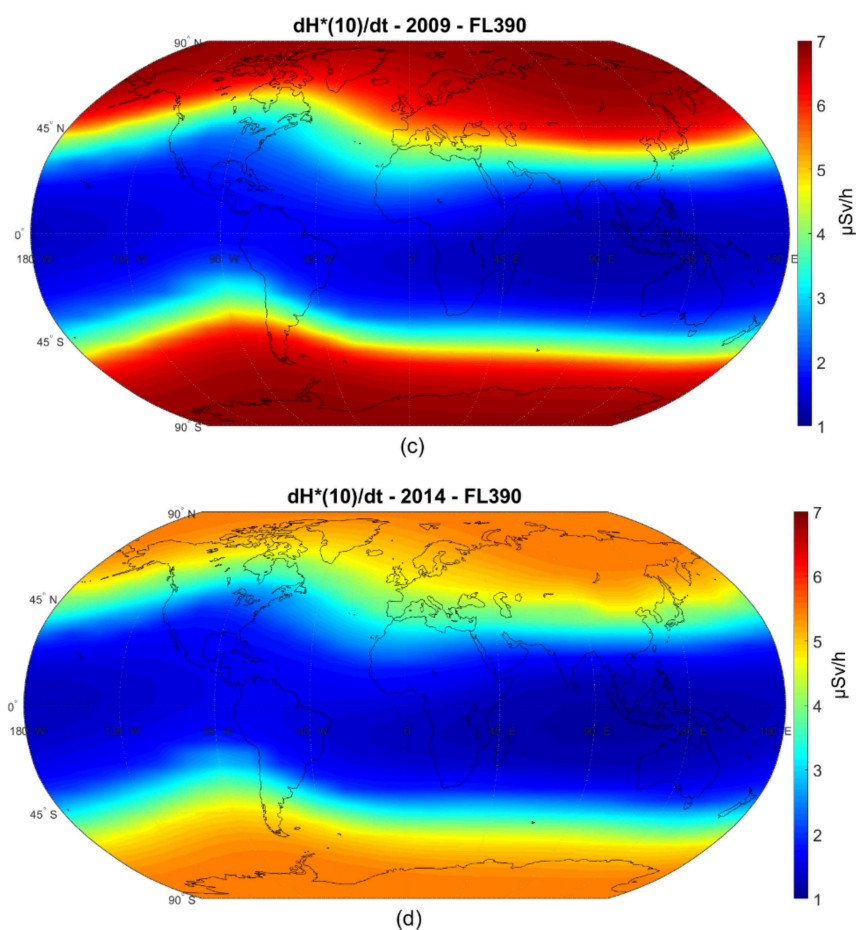

**Figure 2.** Maps of the estimated ambient dose equivalent rate (µSv/h) at FL390: (**a**) during the minimum of solar cycle 23; (**b**) during the maximum of solar cycle 23; (**c**) during the minimum of solar cycle 24; (**d**) during the maximum of solar cycle 24.

Time profiles of the yearly values of the CRII and the ambient dose equivalent rate from the year 1996 to the year 2019 (covering the last two solar cycles) are presented in Figures 3 and 4. Four different geomagnetic cut-off rigidity values were selected: 0.1 GV for the polar region (Figure 3a), 3.1 GV (Figure 3b), 8.5 GV, which corresponded to the middle geographic latitudes and specifically Athens, Greece (Figure 4a) and 14.9 GV for the equatorial region (Figure 4b). The results are provided for the three different atmospheric altitudes which corresponded to the usual flight levels of the commercial aircraft: FL310 (9.45 km a.s.l.), FL350 (10.67 km a.s.l.) and FL390 (11.89 km a.s.l.).

The CRII and dH*(10)/dt values for this time period and for all flight levels can be found in the Supplementary Material. It is interesting that both the CRII (left axis, blue lines) and ambient dose equivalent rate (right axis, red lines) followed a long-term modulation, specifically an 11-year one, at all the aforementioned locations, the same way the GCR intensity did [59–61], since the radiation exposure of aircraft crews was directly linked to the intensity of the cosmic radiation. Furthermore, comparing the time profiles of the three different FLs, it is obvious that the higher the aircraft flew, the higher the CRII and the estimated ambient dose equivalent rates were, since the shielding effect of the atmosphere was reduced and thus the radiation exposure of the aircrew and frequent flyers was higher. It was also observed that the difference among the values at the three FLs was greater as one reached lower rigidities, e.g., polar regions, and became smaller as one reached higher rigidities, e.g., equatorial regions. Since the magnetic field was weaker and more permeable in the polar regions, it allowed even primary cosmic ray particles of lower energies to reach the surface of the Earth, resulting in higher levels of cosmic radiation, unlike in the lower

geographic latitudes where the magnetic lines were almost parallel to the Earth's surface, and therefore provided effective shielding.

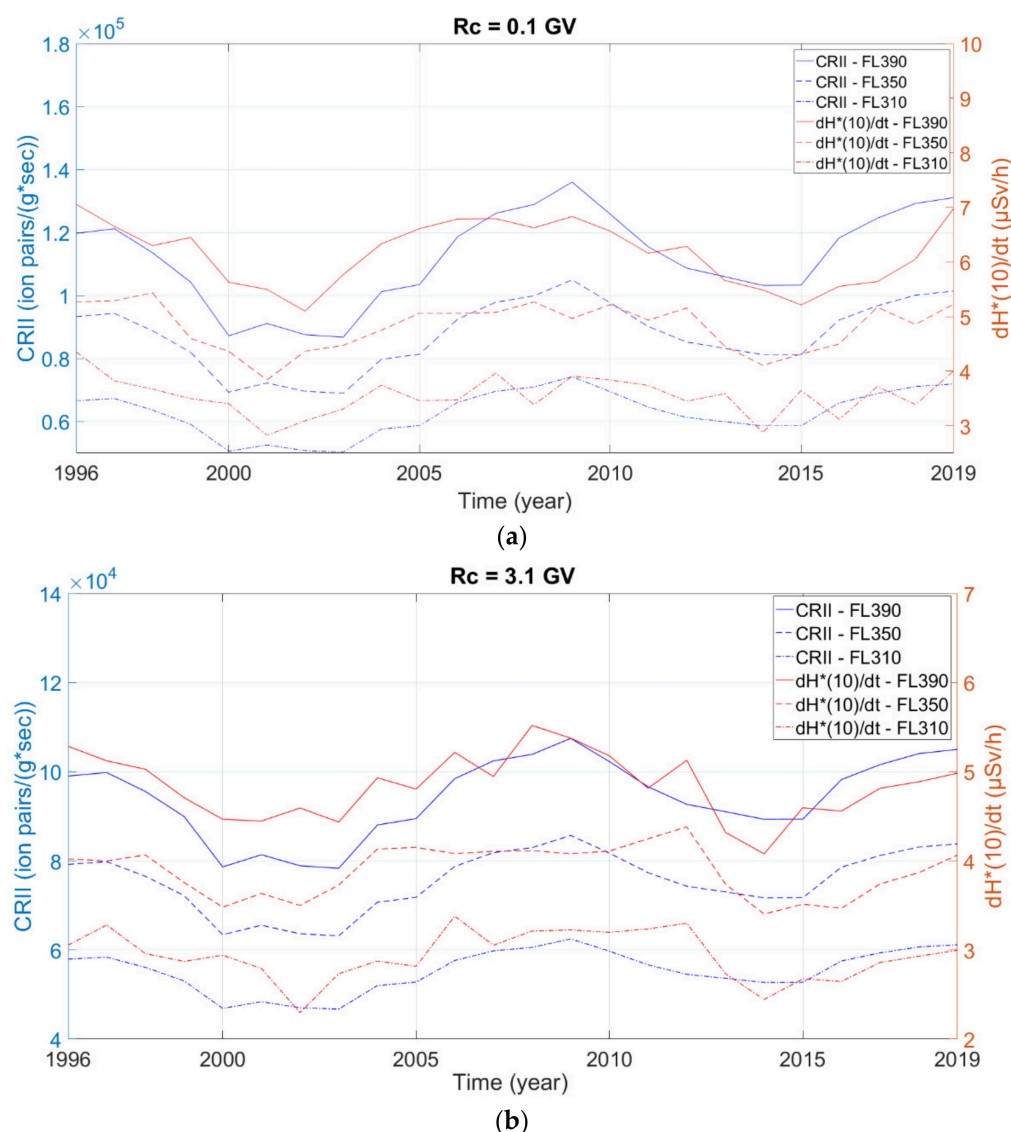

**Figure 3.** Yearly distribution of CRII rate (left axis, blue lines) and ambient dose equivalent rate (right axis, red lines) at three different flight levels (FL310, FL350, FL390), for the time period 1996–2019: (**a**) at a polar region with cut-off rigidity 0.1 GV; (**b**) a region with cut-off rigidity 3.1 GV.

Concerning the CRII, once again, it is noted that during all phases of solar cycle 24, which was a less active solar cycle, the values were greater than those of the respective phases of solar cycle 23, when the solar activity was intense. However, this difference became very small as we moved towards the equatorial regions, which showed that the solar activity mostly affected low-rigidity regions.

More precisely, the CRII decreased by 5.6% near the poles and 24.2% near the equator during SC23 and by 5.4% and 19.1% during SC24, respectively. Similarly, the dependence of the dH*(10)/dt on the solar cycle was most evident near the poles (Rc = 0.1 GV) and, to a lesser extent, near the equator (Rc = 14.9 GV), due to the geomagnetic field shielding, which reflected particles of lower energies. In addition, the dH*(10)/dt decreased by 3.3% near the poles and 19.6% near the equator during SC23 and by 4.1% and 22% during SC24, respectively.

Finally, the correlation between the yearly distribution of the CRII and the estimated ambient dose equivalent rate for all four rigidities mentioned above (0.1 GV, 3.1 GV, 8.5 GV

and 14.9 GV), for all three FLs (FL310, FL350 and FL390), from 1996 to 2019, is illustrated in Figure 5. It is of great importance that a positive correlation between the two physical quantities was observed, with the correlation coefficient being $R^2 = 0.97$. This confirms that the cosmic-ray-induced ionization of the Earth's magnetosphere contributed to the radiation deposited at different locations and altitudes. The data of Figures 3 and 4 are given as Supplementary Material.

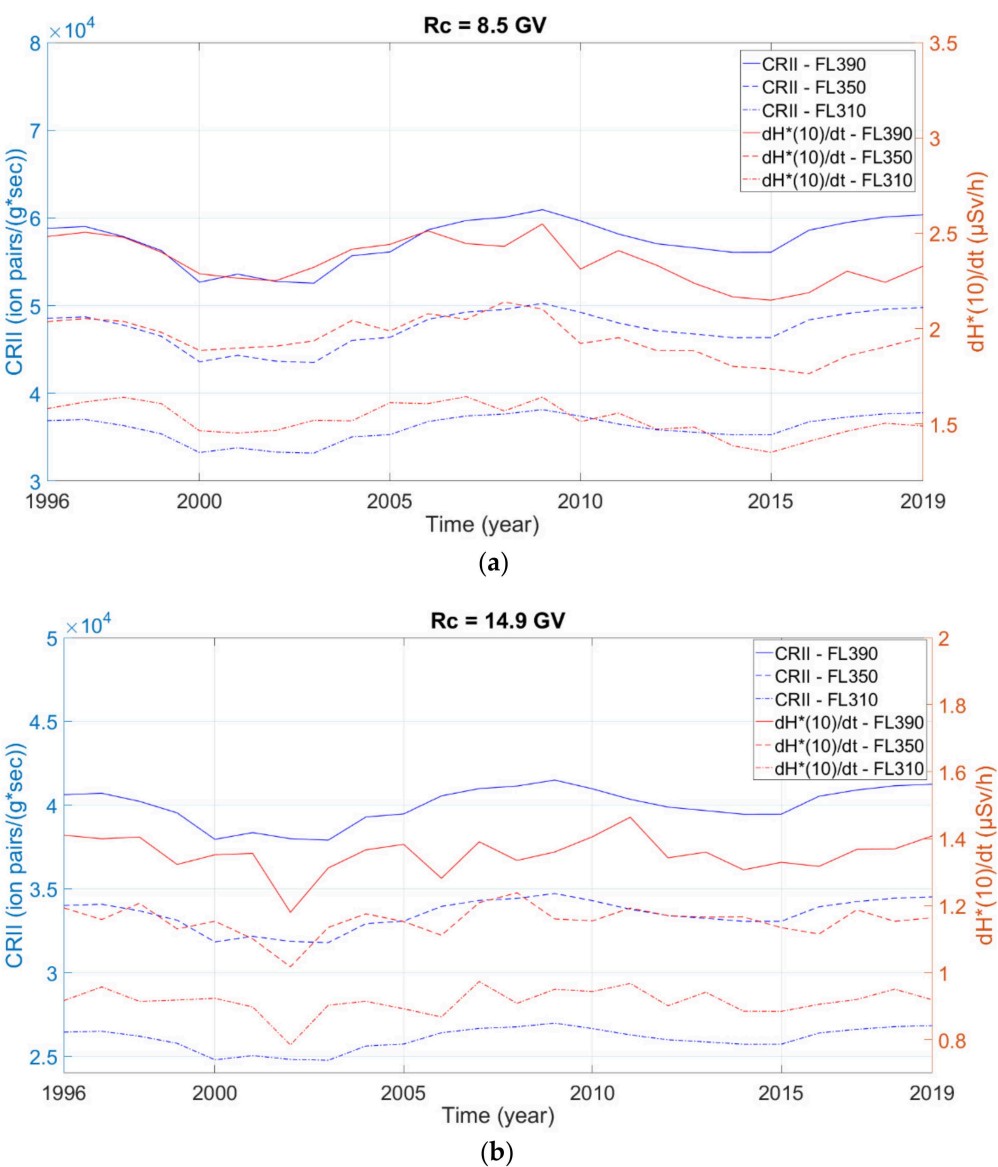

**Figure 4.** Yearly distribution of CRII rate (left axis, blue lines) and ambient dose equivalent rate (right axis, red lines) at three different flight levels (FL310, FL350, FL390), for the time period 1996–2019: (**a**) at a region with cut-off rigidity 8.5 GV; (**b**) an equatorial region with cut-off rigidity 14.9 GV.

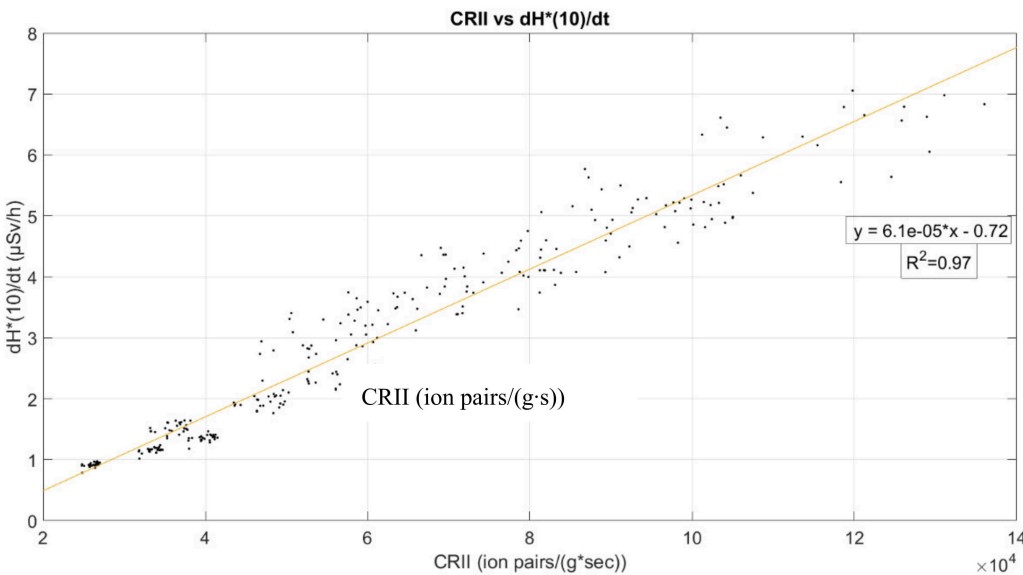

**Figure 5.** Scatter plot of the yearly distribution of the CRII and the ambient dose equivalent rate for the time period 1996–2019 for FL310, FL350, FL390.

### 4. Discussion and Conclusions

In this study, the estimated CRII rate and ambient dose equivalent rate distributions all over the globe were presented for the most common commercial flight levels (FL310, FL350 and FL390) during the recent two solar cycles 23 and 24 (1996–2019). For the calculation of the CRII and the dH*(10)/dt, the CRAC:CRII model of Oulu University [12,13] and the DYASTIMA/DYASTIMA-R software [39,40] were used, respectively. The distribution of both physical quantities was initially depicted in the maps, where the values during the minima and maxima of solar cycles 23 and 24 (for FL390) were illustrated for the entire Earth and all the cut-off rigidities (0–17 GV). The maximum values were observed during the solar minima and at polar regions (approximately 7 μSv/h at FL390), while the minimum values during the solar maxima and at the equatorial regions were approximately 1.2 μSv/h at FL390, due to the anticipated anticorrelation of the cosmic ray intensity with the solar activity, as well as due to the shielding of the geomagnetic field. Furthermore, a comparison between solar activity and ionization can be presented. Using the average sunspot number (ASN) as a measure of solar activity (8 for 1996, 180.3 for 2001, 8.4 for 2009 and 114 for 2014), we can get the following ratios: $ASN_{2001}/ASN_{2014} = 1.58$ and $ASN_{1996}/ASN_{2009} = 0.95$. By calculating the corresponding rations for the maximum values of CRII at FL390 for the same timestamps, we get the following results: $CRII_{2001}/CRII_{2014} = 0.83$ and $CRII_{1996}/CRII_{2009} = 0.88$. According to these results, it is clear that as the solar activity increased, the ionization decreased as expected.

Additionally, different dynamics were observed between solar cycles 23 and 24, due to the difference in solar activity during these two solar cycles which is indicated in Table 1. Concerning the different FLs, it can be concluded that the higher the FL, the higher the CRII and the radiation exposure of aircrews and frequent flyers, since the provided shielding of the atmosphere was reduced in higher atmospheric altitudes. Comparing the CRII and dH*(10)/dt calculations for the four different rigidities/latitudes (0.1 GV, 3.1 GV, 8.5 GV and 14.9 GV), for all three FLs during the entire period 1996–2019, it was noted that the correlation was positive, with the correlation coefficient $R^2 = 0.97$.

From the above analysis, we conclude that both tools gave significant results and can be used in order to study the effect of the ionization and radiation induced by cosmic rays on the environment, space weather, climate change [62,63] and human health [15,22].

Specifically, advances in technology during the last few decades have made air traveling more accessible to everyone. This has led to an increase in the number of flights as well

as an increase in flight altitude, since commercial aircraft are obliged to travel at higher altitudes due to elevated air traffic. The tools mentioned above are of great importance for the assessment of the health effects of the occupational exposure of aviation crews to radiation due to the permanent galactic radiation background. They are also useful for assessing the health effects of potential additional exposure due to sporadic events which cause elevated radiation (such as SEPs and GLEs) [24,64] and radiation clouds (which are regional radiation enhancements possibly due to photons, GCR and outer-belt relativistic electrons), where the ambient dose equivalent rates are significantly increased [65].

Furthermore, all the extracted results have a great impact on ensuring properly shielded avionic electronic systems, which depends on the levels of ionization and radiation. High ionization levels may cause severe malfunctions in semiconductor parts, decreasing the performance and the reliability of the electronic systems. An accurate estimation of such levels would help prevent hardware failures and software errors.

A detailed study of extreme events may contribute to the updating of safety measures and regulations, as well as to the updating of air traffic flow and capacity management, by taking into consideration the respective occupational exposure conditions. An extension of this work is planned in order to include more scenarios, i.e., different input parameters regarding the spectrum of incoming particles based on experimental data (such as the ones provided in [5,6]), more FLs, actual flight plans and extreme events such as SEPs and GLEs.

**Supplementary Materials:** The following supporting information can be downloaded at: https://www.mdpi.com/article/10.3390/app12115297/s1. Table S1. CRII and dH*(10)/dt values for cut-off rigidities 0.1GV, 3.1GV, 8.5GV and 14.9GV over the years 1996–2019.

**Author Contributions:** Conceptualization, P.M. and A.T.; data curation, P.M., A.T. and A.N.S.; formal analysis, A.N.S.; investigation, P.M. and A.T.; methodology, M.G.; resources, P.M., A.T. and A.N.S.; software, P.P. and I.G.U.; supervision, H.M.; project administration, P.M. and A.T.; validation, H.M. and P.K.; writing—original draft preparation, P.M., A.T. and A.N.S.; writing—review and editing, H.M., N.C., M.D. and A.G.G. All authors have read and agreed to the published version of the manuscript.

**Funding:** This research received no external funding.

**Institutional Review Board Statement:** Not applicable.

**Informed Consent Statement:** Not applicable.

**Data Availability Statement:** The datasets generated and/or analyzed during the current study are available from the corresponding author on reasonable request.

**Acknowledgments:** This work is supported by the ESA Space Safety Programme's network of space weather service development and pre-operational activities and supported under ESA Contract 4000134036/21/D/MRP, in the context of the Space Radiation Expert Service Centre. The European Neutron Monitor Services research is funded by the ESA SSA SN IV-3 Tender: RFQ/3-13556/12/D/MRP. A.Ne.Mo.S is supported by the Special Research Account of Athens University (70/4/5803). I.G.U. acknowledges partial support from the Academy of Finland (projects ESPERA No. 321882). Thanks are due to the Special Research Account of the University of Athens for supporting the Cosmic Ray research. Thanks are also due to the Oulu Cosmic ray colleagues for kindly providing cosmic ray data as well as the cosmic-ray-induced ionization model.

**Conflicts of Interest:** The authors declare no conflict of interest.

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
