# Peer review of "Estimation of Cosmic-Ray-Induced Atmospheric Ionization and Radiation at Commercial Aviation Flight Altitudes"

_applsci, doi:10.3390/app12115297_

Round 1

Reviewer 1 Report

The paper covers the topic of cosmic -ray induced radiation and atmospheric ionization during commercial flights. Researches were conducted by means of selected advanced calculation methods. CRII and ambient dose equivalent rate have been estimated at three flight levels. Presented manuscript is really interesting I have only few comments:

Line 50: please expand abbreviation CME

Figure 1: please add units next to the colorbar, not only in description

Line 173: As authors states “it is obvious that CRII has greater…” but can authors provide quantitative estimation of the difference in CRII’s i.e. does the ratio of CRII’s in both cycles correspond somehow to solar activity ratio at the same time?

Figure 2: please add units next to the colorbar

Reviewer 2 Report

The Authors report the study on the ionization of the Earth’s atmosphere due to cosmic ray from the Earth surface to the upper limit of the atmosphere. The reported results have been carried out by using Monte Carlo simulations. The reported results are very interesting. I suggest the minor revision of the manuscript. See the following comments:

  • In the Introduction Section, the overview of the effects on the devices for Space applications should be enlarged (line 84), reporting the impact of the radiations on semiconductor-based devices (see “Measured radiation effects on InGaAsP/InP ring resonators for space applications”, Optics Express, 27(17), 24434-24444 (2019), “Radiation damage of electronic components in space environment,” Nucl. Instrum. Methods Phys. Res., Sect. A 514(1-3), 112–116 (2003); “Radiation damage and annealing in 1310-nm InGaAsP/InP lasers for the CMS tracker,” In Photonics for Space Environments VII (International Society for Optics and Photonics, 2000), 4134, pp. 176–185). Moreover, the Authors should report more details on the Space environment, and its main risks on the mission success (see “The PAMELA experiment on satellite and its capability in cosmic rays measurements,” Nucl. Instrum. Methods Phys. Res., Sect. A 478(1-2), 114–118 (2002).)
  • In order to help the reader to understand the problems related to ionizations, the main physical effects should be reported.
  • The study results very interesting. In order to help the reader to rate the proposed approach, the Authors could compare the achieved results with the ones of Space missions oriented to the cosmic ray detection (as PAMELA experiment), if it is possible for the altitudes of interest. 
  • The quality of the figure should be improved. The axis labels are not clear.
